# Comprehensive Outcomes Affected by Antimicrobial Metaphylaxis of Feedlot Calves at Medium-Risk for Bovine Respiratory Disease from a Randomized Controlled Trial

**DOI:** 10.3390/vetsci10020067

**Published:** 2023-01-17

**Authors:** Lucas M. Horton, Brandon E. Depenbusch, Diana M. Dewsbury, Taylor B. McAtee, Nick B. Betts, David G. Renter

**Affiliations:** 1The Center for Outcomes Research and Epidemiology, Department of Diagnostic Medicine and Pathobiology, Kansas State University College of Veterinary Medicine, Manhattan, KS 66506, USA; 2Innovative Livestock Services, Inc., Great Bend, KS 67530, USA; 3Elanco Animal Health, Greenfield, IN 46140, USA

**Keywords:** antimicrobial use, beef cattle, bovine respiratory disease, economics, feedlot, medium-risk, metaphylaxis, sustainability, welfare

## Abstract

**Simple Summary:**

Bovine respiratory disease (BRD) is the most impactful health disorder in the cattle industry. Metaphylaxis, administration of an antimicrobial to a group of animals at risk for BRD, is a proven method for reducing morbidity and mortality in at-risk populations. However, judicious antimicrobial use is warranted. Medium-risk cattle are a lesser studied population (versus high-risk), where advantages and disadvantages of metaphylaxis are less known, and thus there is more uncertainty on whether or not it should be used. Our objectives were to evaluate the effects of metaphylaxis in a medium-risk population on a comprehensive set of outcomes of value to stakeholders, to estimate the costs and benefits of using or not using metaphylaxis. Using a pull-and-treat program in lieu of metaphylaxis for BRD resulted in substantially less antimicrobial use. However, metaphylaxis improved animal health, performance, and estimated greenhouse gas emissions. While antimicrobial metaphylaxis could be removed from medium-risk populations, there are likely costs associated with such an action that would have negative impacts on animal wellbeing, beef production, economics, and emissions. The framework of values discussed herein should be fully considered by stakeholders considering antimicrobial use decisions.

**Abstract:**

The objectives were to evaluate the effects of metaphylaxis (META) and pull-and-treat (PT) programs on health, antimicrobial use, beef production, economics, and greenhouse gas emissions in cattle at medium risk for bovine respiratory disease (BRD). A randomized complete block design was used at two US commercial feedlots. Steers and heifers [2366 total; 261 (±11.0) kg initial weight] were blocked by sex and feedlot arrival, and allocated to one of two pens within a block (16 pens total, eight blocks). Pens were randomly assigned to treatment: META, tulathromycin injection at initial processing; or PT, tulathromycin injection only for first clinical BRD treatment. Data were analyzed with linear and generalized linear mixed models. There was greater BRD morbidity in PT than META cattle (17.2% vs. 7.3% respectively; *p* < 0.01), and greater total mortality (2.5% vs. 1.1% respectively; *p* = 0.03). Per animal enrolled, 1.1 antimicrobial doses were used for META compared to 0.2 for PT (*p* < 0.01). Per animal enrolled, final live (*p* = 0.04) and carcass (*p* = 0.08) weights were greater for META than PT; however, net returns ($/animal) were not significantly different (*p* = 0.71). Compared to PT, total lifetime estimated CO_2_ equivalent emissions from production were reduced by 2% per unit of live weight for META (*p* = 0.09). While antimicrobial use was reduced with PT, there may be substantial negative impacts on other outcomes if META was not used in this type of cattle population.

## 1. Introduction

Bovine respiratory disease (BRD) is a multifactorial disease complex that has been well established as the most impactful health syndrome in the beef production industry; it is proportionally the greatest contributor to feedlot morbidity and mortality [1,2]. Negative health effects of BRD in turn result in poor cattle performance and substantial economic impacts on the industry [3,4,5,6]. Metaphylaxis is known as the mass-medication of a population where the disease of interest is present, and all individuals are treated [7]. The use of antimicrobial metaphylaxis as a control measure for BRD has been researched extensively [7,8,9,10]. While antimicrobial metaphylaxis is an effective method for curbing BRD incidence rates, scrutiny over the use of antimicrobials in the livestock sector has increased [11], in particular for compounds critically important for human medicine due to concerns of antimicrobial resistance [12].

Generally, the decision on whether or not to use metaphylaxis is determined by estimating the overall BRD risk in a population, where most commonly it is used in cohorts considered at high risk for BRD [8,10]. There are a multitude of factors that are incorporated into estimating risk levels, which may include but are not limited to: calf age, weaning status, weight, weather, distance transported, commingling, and health history [13,14,15,16,17,18]. The vast majority of research has focused on high-risk cattle, for which antimicrobial metaphylaxis is highly effective for reducing BRD morbidity and mortality [9]. A much less studied cattle population that is not well defined in the literature are those at medium (or moderate) risk for BRD; these are cattle that cannot be easily classified as low- or high-risk and therefore fall in between [19]. For producers or veterinarians, there is often an unknown element as to whether or not metaphylaxis should be used for medium-risk cattle due to variable health outcomes. As further evidence to this point, a survey of feedlot nutritionists estimated that 39% of clients use on-arrival metaphylaxis for medium-risk cattle, in contrast to 83% for high-risk and 6% for low-risk classes [20].

Medium-risk cattle were chosen as the target population for this research, due to the knowledge gap surrounding the population and the unquantified advantages or disadvantages of metaphylaxis compared to a pull-and-treat BRD health program. Additionally, the bulk of research comparing antimicrobial metaphylaxis to alternatives has used different antimicrobials, either as different antimicrobial compounds used for metaphylaxis, or using different compounds for metaphylaxis-treated cattle than what is used by controls [9]. Therefore, for a more equal comparison between BRD health programs, it was desired to use identical antimicrobial regimens between programs, with the exception of when the first antimicrobial dose was administered. As there are likely a multitude of residual effects (pros and cons) from deciding to use (or not use) antimicrobial metaphylaxis that extend beyond cattle health and antimicrobial use, an outcomes research approach [21] was used to assess a wide range of stakeholder values. Outcomes evaluated herein include animal health and wellbeing, antimicrobial use, beef production characteristics, economics, and greenhouse gas emissions, all of which affect the overall sustainability of the industry. The goal of the authors was not to deem a health program as superior, as costs and benefits were expected with each program, but rather to construct a framework of values to provide stakeholders with an evidence-based resource to evaluate potential tradeoffs. Thus, the primary objective was to compare metaphylaxis and pull-and-treat health programs using identical antimicrobials on medium-risk cattle morbidity, mortality, and antimicrobial use. Secondary objectives were to evaluate a multitude of critical stakeholder values including cattle performance, carcass characteristics, economics, greenhouse gas emissions, and how they are impacted by the interventions.

## 2. Materials and Methods

As this research was conducted at commercial cattle-feeding facilities, and the treatments and procedures were within the bounds of normal feedlot operations and practices, Institutional Animal Care and Use Committee approval was not acquired. The feedlots followed guidelines from the Care and Use of Agricultural Animals in Research and Teaching [22].

### 2.1. Trial Cattle Population

Crossbred beef steers (*n* = 822) and heifers (*n* = 1544) were received at one of two U.S. commercial feedlots located in Central Kansas (KS) or South-Central Nebraska (NE) between 4 October and 12 December 2019. The specific feedlots were a convenience sample due to their willingness to participate and history with the conduct of randomized controlled trials. Mean initial body weight (BW) of steers and heifers was 264 kg (±9.0 SD) and 259 kg (±12.1 SD), respectively. Cattle were of sale-barn origin, the majority of which were sourced from Oklahoma (*n* = 2190) while one block (described later) of steers was sourced from Alabama (*n* = 176). Cattle were purchased by the same procurement team for both trial locations; i.e., selection and classification of cattle populations was non-differential between the feedlots. Cattle sourced for the trial were deemed at medium risk for BRD by the procurement team. This classification was based on a combination of factors including incoming weight, age (estimated 6 to 8 months), sale-barn origin, unknown pre-conditioning or health history, commingling, transportation distance (mean 722 km, range 402 to 1371 km), and historical experience with similar type cattle purchased from the same locations and season. Health expectations for medium-risk feedlot calves have not been well defined in the literature; however, historical feedlot records indicated that this population was expected to have BRD incidence of approximately 20 to 25% in the control group (no metaphylactic antimicrobial injection at initial processing).

### 2.2. Treatment Structure and Experimental Design

The trial was conducted with a one-way treatment structure in a randomized complete block design. There were two treatments defined by differing health management strategies for BRD, which were pull-and-treat (PT), or antimicrobial metaphylaxis (META) administered at initial processing. Specific antimicrobials used for BRD were identical in their type, sequence, and post-metaphylactic (PMI) and post-treatment (PTI) intervals (Appendix A Table A1). Cattle in the PT group were administered tulathromycin (Draxxin; Zoetis Animal Health, Parsippany, NJ, USA) for first-time clinical BRD treatment, florfenicol (Resflor Gold; Merck Animal Health, De Soto, KS, USA) for second-time clinical BRD treatment, oxytetracycline (Bio-Mycin 200; Boehringer Ingelheim, St. Joseph, MO, USA) for third-time clinical BRD treatment, and danofloxacin (Advocin; Zoetis Animal Health, Parsippany, NJ, USA) for fourth-time clinical BRD treatment, at which point animals were deemed chronic and removed from the trial. All cattle in the META group were administered tulathromycin at initial processing (start of trial); florfenicol was administered for first-time clinical BRD treatment, oxytetracycline for second-time clinical BRD treatment, and danofloxacin for third-time clinical BRD treatment, at which point animals were deemed chronic and removed from the trial. Post-metaphylactic and PTI for tulathromycin, florfenicol, and oxytetracycline were five, three, and two days, respectively. Antimicrobial injections were administered per manufacturer labels (tulathromycin = 1.1 mL/45.4 kg (100 lb), florfenicol = 6.0 mL/45.4 kg, oxytetracycline = 4.5 mL/45.4 kg, danofloxacin = 2.0 mL/45.4 kg). The case definition for animals requiring treatment for BRD is described later.

Within feedlot, cattle were blocked by sex, source, and arrival date. A block consisted of two adjacent pens where cattle were managed identically except for the BRD treatment factor. Once enough cattle were received to fill a block, animals were systematically allocated (by B.D., D.D., and feedlot personnel) to one of the pair of pens by sorting five animals at a time, alternating between each pen, until the desired number of animals per pen was reached. Treatment assignments of pens were concealed at the time of animal allocation to pens. Pens were then randomly assigned to treatment using the RAND function of Excel (Microsoft, Redmond, WA, USA; by B.D.). Pen was therefore considered the experimental unit due to lack of independence of animals within pens. The mean number of animals per pen was 148 (range 81 to 181). Original sample size estimates using α = 0.05, β = 0.20, and historical baseline data from commercial feedlot pens receiving antimicrobial metaphylaxis indicated 12 blocks were necessary to demonstrate 50% reductions in BRD morbidity and mortality, as well as 50% more antimicrobial doses (AMD) used in META versus PT pens. However, due to cattle procurement constraints driven by pen availability this could not be achieved, so calculations were revised using α = 0.10. Thus, a smaller than originally planned sample size was ultimately used. There were a total of six pens enrolled at the KS feedlot (one block steers, two blocks heifers) and 10 pens enrolled at the NE feedlot (two blocks steers, three blocks heifers), for a total of eight blocks (treatment replications).

### 2.3. Cattle Management

#### 2.3.1. Feedlot Receiving and Processing

At the time of arrival to the feedlot, animals were penned separately by sex, origin, and arrival date, and were given ad libitum access to long-stemmed hay and water with a minimum 24 h rest period prior to initial processing. On the morning of trial allocation of a block, cattle were sorted into trial pens (as described above) prior to initial processing. Once pens were sorted, cattle were weighed in groups on a platform scale for determination of total and mean initial BW of the pen. Each pen within a block was then processed as a group by a custom processing crew. Tulathromycin was administered via subcutaneous injection to META pens at the manufacturer-labeled dose. No form of alternative (e.g., saline) injection was administered to PT cattle, therefore processing personnel were not blinded to assigned treatments; however, no measurements were taken by processors, nor were trial cattle differentiated from other non-trial cattle processed the same day at the feedlots. All other procedures and products administered were identical between pens within a block; however, they were allowed to vary between blocks and feedlots. Differences between feedlots and blocks were due to differences in sex, BW, product availability, and consulting veterinarian philosophy.

At initial processing, all animals received two ear-tags containing a lot number and unique animal identification number; a four-way modified live antiviral vaccine (Pyramid-4; Boehringer Ingelheim) for bovine viral diarrhea type 1, infectious bovine rhinotracheitis, parainfluenza 3, and bovine respiratory syncytial virus; a subcutaneous parasiticide injection (Dectomax; Zoetis Animal Health, Parsippany, NJ, USA); and an oral anthelmintic (Safeguard; Merck Animal Health, De Soto, KS, USA).

At initial processing at the NE feedlot, steer blocks were administered a coated, extended release hormonal implant (Revalor-XS; Merck Animal Health, De Soto, KS, USA), and heifer blocks were administered an uncoated implant (Revalor-IH; Merck Animal Health, De Soto, KS, USA). After approximately 60 days on feed (DOF), NE heifer blocks were re-implanted with an uncoated implant (Revalor-H; Merck Animal Health, De Soto, KS, USA), and also administered a clostridial vaccine (Vision 7 with SPUR; Merck Animal Health, De Soto, KS, USA), a modified live vaccine for control of bovine rhinotracheitis virus and *Leptospira pomona* (Titanium IBR LP; Elanco Animal Health, Greenfield, IN, USA), and a topical insecticide (Embargo; MWI Animal Health, Boise, ID, USA).

Cattle at the KS feedlot were re-vaccinated 13 to 14 days following initial processing, where all received a Pyramid-4 booster, an autogenous foot-rot vaccine (Newport Laboratories, Inc., Worthington, MN, USA), and a topical insecticide (Clean-Up II; Elanco Animal Health, Greenland, IN, USA). The steer block was administered an uncoated implant (Revalor-IS; Merck Animal Health, De Soto, KS, USA) and heifer blocks were administered Revalor-IH. After 77 DOF, the KS steer block was re-implanted with Revalor-IS, and also received a modified live vaccine for bovine rhinotracheitis virus and bovine viral diarrhea virus Types 1 and 2 (Bovi-Shield GOLD IBR-BVD; Elanco Animal Health, Greenland, IN, USA), an autogenous foot-rot vaccine (Newport Laboratories, Inc., Worthington, MN, USA), and Clean-Up II.

At approximately 120 DOF, all cattle received a terminal implant (Revalor-200; Merck Animal Health). All NE blocks also received a clostridial vaccine (Vision CD-T with SPUR; Merck Animal Health), Titanium IBR LP, and Embargo. Animals at the KS feedlot received an autogenous foot-rot vaccine (Newport Laboratories, Inc., Worthington, MN, USA), Bovi-Shield GOLD IBR-BVD, and Clean-Up II, except for the last block enrolled in the study (heifers) which did not receive Clean-Up II. All re-vaccination, re-implant, and terminal implant procedures were performed by a custom processing crew who were blinded to experimental treatments. The reader is referred to Appendix A Table A2 for an alternative description of pharmaceutical products administered at each processing event for each block.

#### 2.3.2. Animal Feeding

Cattle were housed in dirt-surfaced pens equipped with automatic waterers to provide *ad libitum* access to well water. After trial enrollment, cattle were fed a starter diet and gradually transitioned to a finishing diet over an average of 39 days (range 25 to 49) for ruminal adaptation to high-starch inclusion, and for controlling cattle growth rate due to light initial BW. A series of “steps” to achieve intermediate concentrations of starter and finishing diets were used for the transition. Ingredient and calculated nutrient compositions of starter and finishing diets for each feedlot are in Table 1. Finishing diets were formulated to meet all nutrient requirements specified in the Nutrient Requirements of Beef Cattle [23]. Cattle were fed thrice daily beginning at 0700 h, intending to provide *ad libitum* consumption. All cattle received ractopamine hydrochloride (250 mg/animal/day; Zoetis Animal Health, Parsippany, NJ, USA) for their final 28 to 42 DOF, with the exception of the last block of heifers that were in enrolled in the trial (KS feedlot). The variation in days fed ractopamine (and not fed to the last block) was due to uncertainty around when the block could be harvested due to reduced abattoir slaughter capacity due to COVID-19-induced labor and personnel restrictions. Daily feed deliveries were summed at the pen level over the trial period on a dry matter basis, and divided by animal-days to estimate mean daily dry matter intake (DMI) per animal. Animal-days is the total number of days cattle were in the pen multiplied by the number of cattle, which includes animals removed from the trial and mortalities, up until the day of removal or death.

#### 2.3.3. Animal Health

Following trial enrollment, META pens were ineligible to have animals pulled and treated for BRD for five days due to the five-day tulathromycin PMI. Animals were eligible to be pulled for any non-respiratory disorders. Trained cattle caretakers were not blinded at this time as differentiation between PT and META pens were required due to labor constraints. Animals were eligible to be treated for BRD when clinical signs were observed given that the animal was outside of any PMI/PTI windows. Clinical signs of BRD were labored breathing, depression, anorexia, coughing, and nasal discharge. While additional chute-side diagnostics were taken including BW, lung-score, and rectal temperature, they did not impact whether or not animals were treated; BRD treatments were made exclusively on clinical signs, determined by veterinarian-trained assessors. Antimicrobial regimens for PT and META (Section 2.2) only applied for the treatment of clinical BRD, and an intent-to-treat analysis was used for evaluation of these outcomes. The different antimicrobial compounds used cannot be considered equivalent in type nor dosage (i.e., 1 mg/mL tulathromycin ≠ 1 mg/mL florfenicol). Therefore, evaluation of antimicrobial use for the treatment of BRD was considered using antimicrobial doses (AMD) as the primary metric. Antimicrobial doses are the total count of antimicrobial injections for BRD that an animal received, which was collapsed at the pen level for total AMD per pen. Treatment of other ailments (e.g., lameness, digestive disorders) were not controlled by the trial protocol, and were treated per feedlot standard operating procedures prescribed by the consulting veterinarian using manufacturer labels and guidelines for veterinary products. For all diagnosis types, it was intended that animals be returned to their home pen following treatment if ailments allowed. Animals deemed unable to return home were placed in a hospital pen for recovery where they were allowed to reside for a maximum of four days. If on the fifth day an animal was still unable to return home, it was removed from the trial. Animals that were treated four times for any reason (or META cattle treated three times for BRD) were deemed chronic and removed from the trial, where date, BW, and final reason for removal were recorded. In addition to monitoring health, pens were inspected daily for mortalities; date of occurrence, pen, animal identification, and diagnosis of death were recorded. If diagnoses were not obviously apparent, a necropsy was performed by trained personnel for final determination. All treatment, removal, and mortality records were collapsed at the pen level.

### 2.4. Cattle Harvest

Final weights and harvest occurred on the same day for both pens (treatments) within a block. The mean DOF was 218 (range 187 to 233). On the final day of the trial for each block, cattle were weighed in groups on a platform scale for determination of total and mean final BW for each pen. A 4% BW shrink was applied to final weights which were used for final calculations of average daily gain (ADG) and gain:feed. These final calculations were made on the basis of dead and removed animals included or excluded. Cattle were then loaded on trucks and transported (KS, 233 km; NE, 60 or 415 km) to a commercial abattoir. Blocks were harvested between 8 April and 23 July 2020. At the packing plant, individual animal hot carcass weight (HCW) was measured on the day of harvest, while camera system measurements of ribeye area, 12th-rib fat thickness, marbling score, calculated yield grade (YG), United States Department of Agriculture (USDA) YG, and USDA Quality Grade (QG) were made following a minimum 24 h carcass chill period. Abattoir personnel responsible for harvesting and recording carcass measurements were blinded to treatments. Hot carcass weight, ribeye area, 12th-rib fat, marbling score, and calculated YG were averaged to create pen-level means. Records of USDA YG and QG remained at the carcass level for analysis. Percent dressed yield was calculated by dividing mean HCW by mean shrunk final BW.

### 2.5. Economic Assessment

A partial budget using prices reflective of the time of the trial conducted was constructed to estimate economic implications of the treatment interventions. All components of the budget used input prices for cost and revenue that were multiplied by pen-level outcomes. Cost parameters in the partial budget were: cattle purchases, processing, BRD antimicrobials, non-BRD treatments, rendering fees (for mortalities), feed, and yardage (accounts for labor and facilities maintenance). Revenue from cattle sales was considered on the basis of two sale types, live (non-adjusted) sales, or dressed sales adjusted for carcass quality-based premiums and discounts. Revenue parameters in the partial budget were: sale of animals removed (culled) from the trial, live cattle sales (if sold on a live basis), and (if sold on a dressed basis) dressed cattle sales adjusted for YG, QG, and over- or under-weight carcasses. The final outcome was net return, defined as total pen revenue minus total pen costs, per animal, on a live or dressed sale basis.

Cattle purchase prices were $154.01/45.4 kg (100 lb) for steers and $138.81/45.4 kg for heifers, which were multiplied by the total initial BW of each pen. These prices were the average of weekly prices from the range of dates that animals were enrolled. These prices were sourced from the Livestock Marketing Information Center (LMIC [24]), Oklahoma City Auction, for medium and large frame #1 feeder steers and heifers weighing 249 to 272 kg (550 to 600 lb; contains the mean weight of both sexes enrolled in the trial). Monthly prices for dry feed, feed markup, and yardage from the Central Plains region [25] were averaged (weighted by the months pens were on trial), resulting in a mean price of $254.14/907 kg (2000 lb) dry feed, which was multiplied by the total feed consumed by each pen. Actual prices reported by the feedlots were used for products and chute-charges (labor and equipment) from each processing event (Section 2.3.1). A mean total processing charge of $17.38/animal was made, and multiplied by the number of animals enrolled in each pen; cost of tulathromycin given to META cattle was excluded from this charge. Similarly, reported prices from the feedlots for BRD antimicrobials were used: tulathromycin = $3.95/mL, florfenicol = $0.56/mL, oxytetracycline = $0.10/mL, and danofloxacin = $1.67/mL. The price of tulathromycin was multiplied by the labeled dose (1.1 mL/45.4 kg) and by the total initial BW of META pens for total metaphylaxis cost. Clinical BRD treatments were handled similarly; the price of florfenicol, oxytetracycline, and danofloxacin were multiplied by their respective dosages (Section 2.2) and the actual BW of META cattle treated one, two, or three times for BRD, respectively. Prices of tulathromycin, florfenicol, oxytetracycline, and danofloxacin were multiplied by their respective dosages and by the actual BW of PT cattle treated one, two, three, or four times. Additionally, for every animal treated for clinical BRD, a $1.50 chute-charge was applied. As non-BRD treatments were not controlled for by the trial design, a constant mean price by diagnosis category was applied from the 2013 USDA—National Animal Health Monitoring System (NAHMS) report [26], which was similar to prices used at the time by the feedlots. Cases of non-BRD respiratory disease (primarily acute interstitial pneumonia; AIP) were charged $21.70/treatment; cases of musculoskeletal/trauma were charged $13.40/treatment; and (due to infrequency) all other cases were combined in a single “other” category, which were charged $12.30/treatment, which is the average of “digestive problems,” “bullers,” and “central nervous system problems” from the USDA—NAHMS report [26]. The final cost applied was for mortality, where a $38.24 charge per death was used, representing an estimated cost for rendering a dead animal. This value was based on an original $24.11/render fee from a 2002 report [27]; due to the dated nature of this value, it was inflation-adjusted to the average Producer Price Index for Animal Slaughtering and Processing [28] for the months of trial conducted, which was anecdotally confirmed as reasonable by industry experts.

Revenue from animals removed from the trial was estimated using a relationship described by Horton et al. [29], where a fraction (determined by categories of feedlot cull HCW) of the national dressed Breaker cow (over 227 kg) price is used. Weekly national dressed Breaker cow prices from the USDA—Agricultural Marketing Service (AMS) [30] were averaged based on the range of dates blocks were harvested for a value of $124.57/45.4 kg. The HCW of removed animals was estimated using a regression provided by Tatum et al. [31]. Based on HCW category [29], a proportion of the Breaker cow price was multiplied by the total removed animal HCW from each pen.

The majority of pen revenue was from finished (fed) cattle sales. For the live sale scenario, $103.94/45.4 kg for steers and $104.43/45.4 kg for heifers was multiplied by total shrunk pen BW for estimated revenue from live cattle sales. These values are an average of weekly prices representative of the timeframe pens were harvested for steers or heifers expected to grade over 80% Choice, and sold live (free-on-board) [32]. The base price used for dressed fed cattle sales was the live fed price stated above divided by a common 63.5% dressed yield (i.e., 0.635) resulting in values of $163.69/45.4 kg for steers and $164.46/45.4 kg for heifers. Base dressed revenue was estimated by multiplying the base fed dressed price by total pen HCW. Base dressed revenue was then adjusted for carcass-based premiums and discounts (YG, QG, and weight) by constructing an average price (similar to above) reported by the USDA—AMS [33]. Due to the infrequency of carcasses grading below Select, all sub-Select carcasses received the discount for Standard carcasses. No adjustment was made to Choice or YG 3 HCW, as these categories are set as the base. Premiums for Prime, YG 1, and YG 2 carcasses (per 45.4 kg) were $8.29, $5.43, and $2.42, respectively. Discounts for Select, Standard, YG 4, YG 5, HCW under 249 kg, and HCW over 476 kg (per 45.4 kg) were $(−14.10), $(−29.66), $(−9.80), $(−12.68), $(−24.21), and $(−14.07), respectively. Premium and discount prices were multiplied by the total pen HCW that fell in each category (e.g., Prime price × pen Prime HCW). These additions and subtractions from the base dressed revenue formed total revenue from animals sold on a dressed basis. Revenue from animals removed from the trial was summed with revenue from each sale basis (live or dressed) for the final estimation of total pen revenue. The framework of parameters and prices used in the partial budget is in Appendix A Table A3.

### 2.6. Production Emissions

Estimation of calculated carbon dioxide equivalent (CO_2_e) emissions were modeled using UpLook (Elanco Animal Health), which incorporates methodologies from the USDA [34], the Intergovernmental Panel on Climate Change [35], and the US EPA [36], for calculating greenhouse gas emissions from agricultural practices. All greenhouse gasses evaluated in the model were converted to CO_2_e for interpretation. Incoming calf estimates were assumed, and represent the carbon footprint of animals received for the trial prior to their arrival (namely cow-calf production parameters), for which breed and initial BW were primary factors for the model. Assumptions incorporated enteric methane, manure, fertilizer used for pastures, supplemental feed, as well as on farm fuel and electricity usage (including cattle transport to the feedlot). The incoming calf CO_2_e estimate was incorporated in the final total, to evaluate if treatment interventions implemented in the finishing phase would impact the total lifecycle greenhouse gas footprint associated with beef production. The remaining model inputs were feedlot- and pen-level data from the trial. Impacts of feed were included based on the production of individual ingredients and associated CO_2_, N_2_O, and CH_4_ emissions from cultivation, planting, fertilization, pesticide use, and harvesting using published carbon footprint values for feed ingredients [37,38]. Individual dietary components for each formulation (Table 1) and the total dry feed delivered of each (to pens) were part of this estimate. Data on feedlot energy use were included, which accounted for fuel, natural gas, and electricity by allocating proportionally by animal-days from yearly use. Methane and nitrous oxide from manure, and animal enteric methane were accounted for and incorporated dietary crude protein (from dietary nutrient analysis) and digestible energy (from USDA guidelines [34]), along with the total DMI of each diet. Manure CO_2_e was derived from: manure methane, direct manure N_2_O, indirect manure N_2_O from volatilization, and indirect manure N_2_O from leaching and runoff equations from USDA guidelines [34]. Other critical pen-level components of the models were *n* animals enrolled, total initial BW, animal-days, animal removals and mortalities, *n* final animals, final BW, and HCW. Estimates were adjusted for enteric methane reduction from monensin and steam flaked grain processing based on USDA guidelines [34]. Mature body weights were adjusted for sex, hormonal implant, and beta-agonist use [23]. Differences in carbon emissions from the production of pharmaceuticals used on the treatment groups were *de minimis*. The investigator responsible for execution of these models was blinded to experimental treatments (N.B.). For analysis purposes, estimated CO_2_e from feedlot operations (feed ingredients, commodity trucking, and fuel and utilities) were summed to a single metric, as were all manure components for total manure CO_2_e. Final values were interpreted per animal enrolled, and per unit of beef production (live BW and HCW).

### 2.7. Statistical Analyses

All outcomes were modeled with pen as the experimental unit, treatment as a fixed effect, and block within feedlot as a random intercept; additionally, sex was included as a binary covariate for adjustment of model estimated means. Continuous outcomes (e.g., ADG, HCW) were fit using linear mixed models (Proc GLIMMIX, SAS 9.4; SAS Institute Inc., Cary, NC, USA) with a Gaussian distribution and identity link function using restricted maximum likelihood estimation, a Kenward–Roger adjustment of degrees of freedom and standard errors, and Newton–Raphson with ridging optimization procedures. Visual (graphical) assessments of conditional and marginal studentized residuals were conducted to ensure assumptions of homoscedasticity and normality were reasonably met. Generalized linear mixed models (Proc GLIMMIX, SAS 9.4) were used for outcomes fit with binomial (e.g., morbidity and mortality measures), Poisson or negative binomial (e.g., antimicrobial doses), and multinomial (e.g., QG) distributions, and used Newton–Raphson with ridging optimization procedures. Binomial (logit link) and Poisson [log link with appropriate offset (e.g., animals enrolled, HCW)] regressions were first run with a Laplace approximation for assessment of overdispersion (Pearson-Chi^2^/df > 2). As some outcomes modeled with Poisson regression were over-dispersed, negative binomial models were instead used. After evaluating fit, these models were then re-run to include Kenward–Roger degrees of freedom adjustment and used a residual pseudo-likelihood estimation technique. Estimates were back-transformed to probabilities or counts for interpretation. Ordinal outcomes were analyzed with multinomial regression using a cumulative logit link function, a proportional odds assumption, Kenward–Roger’s degrees of freedom, and residual pseudo-likelihood estimation. Multinomial models also included an additional random intercept for pen to account for lack of independence of animals within pens due to the carcass-level data structure. Potential differences in the distributions of ordered categories of YG and QG between treatments were assessed with a likelihood ratio test. As these models do not allow estimation of model adjusted means for each category, raw frequency statistics are reported. Due to sparseness of carcasses with a QG below Select, they were combined to form a “Select or below” category for model convergence. Due to having a smaller than originally planned sample size, a more liberal significance threshold was used, and set *a priori* at α = 0.10.

## 3. Results

For all outcomes, the number of pens analyzed was consistent with the number enrolled in the trial (*n* = 16).

### 3.1. Animal Health and Antimicrobial Use

Health outcomes pertaining to morbidity, mortality, antimicrobial use, and animals removed from the trial are in Table 2. For reiteration, antimicrobial metaphylaxis used for control of BRD in META pens was not considered a treatment for animal morbidity; only cases of clinically diseased animals were considered a treatment. Indiscriminant of reason for treatment, there were approximately half the amount of first-time treatments (*p* < 0.01), as well as a 42% decrease in the proportion of second-time treatments (*p* = 0.03), for META cattle compared to PT. There was no evidence of a difference between treatments for animals requiring a third-time treatment for any reason (*p* = 0.17). There were no animals in the trial that required to be treated four times. The vast majority of animals requiring treatment in the trial was due to BRD, and the number of total BRD treatments was more than double for PT than for META (*p* < 0.01). There was no evidence of differences between treatment counts in other categories, which were non-BRD respiratory disease (*p* = 0.21; AIP except for one case of diphtheria), musculoskeletal/trauma (*p* = 0.63), or “other” (*p* = 0.86). First-time BRD treatments were reduced by 57% for META versus PT (*p* < 0.01), and those requiring a second BRD treatment were reduced by 49% (*p* = 0.05). There was not a significant difference between treatments for animals requiring three BRD treatments (*p* = 0.21). Considerably more AMD (antimicrobial doses or injections) for BRD were used per animal enrolled (*p* < 0.01) and per 45.4 kg of HCW (*p* < 0.01) for META cattle compared to PT. This can be interpreted as: per 100 animals, 110 AMD were used for META while 21 were used for PT; or per 10,000 kg of HCW, 30 AMD were used for Meta while 6 were used for PT.

Due to the sparseness of data for animals removed from the trial and mortalities for non-BRD reasons, they were combined into a single “other” category. There was no evidence of a difference in the proportion of animals removed from the trial for BRD (*p* = 0.35), other reasons (*p* = 0.21), nor for the total of all removals (*p* = 0.89). While there was no significant difference between treatments for BRD mortality (*p* = 0.34), there was a 61% reduction in “other” mortalities (*p* = 0.04), as well as a 55% reduction in total mortalities (*p* = 0.03) for META compared to PT. Therefore, the estimated attributable risk of using a PT BRD health program in lieu of META was 1.4 excess deaths per 100 animals. The total number of animals that were either removed from the trial or died (fallouts) was reduced by 31% for META compared to PT (*p* = 0.10).

### 3.2. Feedlot Performance and Carcass Characteristics

Outcomes relating to feedlot performance are in Table 3. There was no evidence of a difference in initial BW between treatments (*p* = 0.52). Animals administered META upon arrival consumed approximately 0.19 kg more feed (DMI) per day than PT (*p* = 0.06). On a dead-and-removed-animals-excluded basis, on average, META cattle had 6 kg heavier final BW compared to PT (*p* = 0.10). There was no evidence of differences between treatments for ADG (*p* = 0.13), nor gain:feed (*p* = 0.83). On a dead-and-removed-animals-included basis, which accounts for mortalities and animals removed from the trial, final BW increased by 15 kg (*p* = 0.04), and ADG increased by 0.06 kg (*p* = 0.05) for META cattle compared to PT. Due to the increased DMI observed for META, the improvement in ADG did not translate to an improvement in gain:feed, as there was not a significant difference between treatments (*p* = 0.24).

Table 4 contains results of carcass outcomes. On a per carcass basis, there was no evidence of an HCW difference (*p* = 0.12). However, if considered on a per animal enrolled basis (similar to deads and removals included in Table 3), META cattle had 19 kg heavier HCW on average compared to PT. There was no evidence of differences for any other carcass characteristics (*p*-values ≥ 0.28).

### 3.3. Economic Assessment

Results from a partial budget for the assessment of economic implications between the treatments are in Table 5, where the main budgetary components of interest are shown. As anticipated due to similar initial BW (Table 3), the average purchase price per animal did not significantly differ between treatments (*p* = 0.54). Due to greater DMI, META cattle had a higher cost of feed and yardage than PT by just over $14 per animal enrolled (*p* = 0.06). However, there was no evidence for a difference in cost of gain (*p* = 0.24), which is akin to not observing a treatment effect on gain:feed (Table 3). As every META animal was administered tulathromycin at initial processing, driving increased AMD per animal (Table 2), the cost of antimicrobials used for BRD was more than $21 higher per animal enrolled for META compared to PT (*p* < 0.01). The cost of treating all non-BRD classified ailments did not differ significantly (*p* = 0.72). On a live sale basis, META cattle received nearly $15 more per animal shipped (*p* = 0.10), and (alternatively) nearly $32 more per animal enrolled (*p* = 0.07) compared to PT. Although the treatment differences were fairly similar numerically on a dressed sale basis, there was no evidence of an effect on revenue received per carcass (*p* = 0.21), or (alternatively) per animal enrolled (*p* = 0.12), likely due to the increased variability that occurs when incorporating premiums and discounts. There were no significant net return differences between treatments on either a live or dressed sale basis (*p*-values ≥ 0.68).

### 3.4. Production Emissions

Calculated estimates of CO_2_e emissions from multiple sources associated with the production of trial cattle are shown in Table 6. Due to similar initial BW of the animals between treatments, and identical sources within blocks, there was no evidence of a difference for estimated CO_2_e emissions derived from the pre-feedlot, calf production phase (*p* = 0.55). Estimates of CO_2_e associated with the finishing phase (trial period) indicated that more CO_2_e emissions were generated (per animal enrolled) by META cattle compared to PT for feedlot operations (*p* = 0.06; includes emissions associated with the production of dry feed ingredients, commodity trucking, and fuel and utilities), manure (*p* = 0.09), and enteric methane (*p* = 0.06). These variables combined yielded an estimated 45.9 kg more CO_2_e emitted for META cattle than PT (*p* = 0.07) per animal enrolled. Similar to the concept of feed efficiency (gain:feed), these greater emissions per animal were offset when factoring in the additional beef produced by META pens, as there was no evidence of a difference between treatments in total finishing period CO_2_e per kg of final BW (*p* = 0.94) or HCW (*p* = 0.94). The estimated total lifecycle CO_2_e emissions associated with the production of these cattle, which combines pre-feedlot and feedlot phases, was greater per animal enrolled for META compared to PT (*p* = 0.08), for which the difference can be attributed to the feedlot phase. However, META led to reduced CO_2_e emissions per unit of final BW (*p* = 0.09) and HCW (*p* = 0.10) compared to PT. In other words, when META was used in lieu of PT, total lifecycle emissions were used more efficiently due to the resulting increase in beef production.

## 4. Discussion

While primary objectives of the trial design were to evaluate cattle health and antimicrobial use, an outcomes research approach [21] was used to evaluate a multitude of values of importance to stakeholders. Interventions to manage an important health issue such as BRD can be considered through multiple perspectives; to re-emphasize, the objectives were not to just determine which program had the best impacts on cattle health, but rather to present a framework of values, as interpretations likely vary depending on what is deemed most important to a stakeholder. Critical values emphasized in this research were animal health, antimicrobial use, animal production, economics, and estimated greenhouse gas emissions, which are all important considerations in the broader context of beef industry sustainability. Additionally, the intentions of this research were to have direct comparisons between the health programs using the same antimicrobials, rather than comparing efficacies of different antimicrobials used for metaphylaxis which have been studied in depth [9]. Using identical antimicrobials for BRD treatment regimens as part of the experimental design allowed for a more explicit comparison of the programs themselves, rather than a forced comparison of different antimicrobial compounds. To a degree, there are limited direct comparisons with published literature, as research with a similar methodology is not known by the authors. Therefore, the discussion herein focuses on the interpretation and implications of the two BRD health programs, and their potential impacts on stakeholder values.

Although improvements have been made for classifying and predicting morbidity based on risk factors [15,16,17,18], there are still inherent challenges, and outcomes are still variable. Feedlot cattle populations at medium (or moderate) risk for BRD are not as well defined as low- or high-risk populations; ultimately, they fall somewhere in between, and there may be more variability to be expected around outcomes observed. That was the case in this trial, as overall BRD morbidity was lower than anticipated, but variable amongst blocks. Pen-level BRD morbidity (percent first treatments) ranged from 11.1 to 37.1% for PT, and 2.8 to 14.0% for META. Similar observations were made by Nickell et al. [39] in (negative control) cattle populations described as medium-risk. This reality helps explain why there is uncertainty around whether or not to use antimicrobial metaphylaxis in medium-risk cattle, and is why this population was chosen for the trial, as there is need for more evidence-based resources for decision makers.

Tulathromycin has been established as a top-tier antimicrobial for metaphylactic use for BRD [9]. Therefore, the observed improvements in BRD morbidity were not unexpected, and are consistent with the literature [9,40,41,42]. A 50% reduction in morbidity risk is a common threshold associated with antimicrobial metaphylaxis [10], and is comparable to observed effects herein. Tulathromycin is also considered a top-tier antimicrobial for therapeutic treatment of BRD [43], and although tulathromycin was administered for first-time PT BRD treatments, there was still a greater proportion of PT cattle that required a second clinical BRD treatment versus cattle in the META groups. However, PT cattle treated twice for BRD received two AMD, while META cattle “treated twice” would have received three (i.e., the comparison is not 1:1 for antimicrobials received). Effects of the administered antimicrobials extend beyond BRD; for example, per manufacturer label, in addition to bacterial pathogens associated with BRD, tulathromycin is also indicated for the treatment of infectious bovine keratoconjunctivitis associated with *Moraxella bovis*, and foot-rot associated with *Fusobacterium necrophorum* and *Porphyromonas levii*. One could reasonably presume that antimicrobial effects also extend to conditions beyond label indications; for example, long-acting antimicrobials including tulathromycin and florfenicol may have impacts on infectious lameness per synovial fluid pharmacokinetic measurements [44]. While treatment effects on non-BRD morbidity were not observed (possibly due to low incidence), the proportion of mortalities due to non-BRD (“other”) reasons was greater for PT cattle than META, leading to an overall difference in mortality. This may be due to increased AMU in META leading to an impact on subclinical, non-BRD ailments in animals that would have otherwise died, misclassification of BRD versus non-BRD outcomes, or the overall reduction in BRD morbidity in META cattle may have yielded healthier populations, resulting in reduced risk of death classified as non-BRD. Due to data sparseness leading to issues with model convergence, “other” mortalities could not be broken down by individual disease categories within. For perspective, it appears the mortality difference was not due to a single sub-category, as the number of animal mortalities diagnosed as non-BRD respiratory, musculoskeletal/trauma, digestive, or “other” were: six, four, six, and seven, respectively, for PT; and two, one, four, and two, respectively, for META.

Antimicrobial use for treatment of BRD was a crucial outcome to consider, due to increasing public scrutiny, and pressure to reduce the use of critically important antimicrobials for human medicine [12]. The majority of antimicrobial doses for both treatments were with tulathromycin, a macrolide, and therefore considered critically important. Less used but still important antimicrobials in the trial were florfenicol (an amphenicol), oxytetracycline (a tetracycline), and danofloxacin (a fluoroquinolone), of which the antimicrobial classes of the first compounds are considered highly important, while the latter is considered critically important [12]. Research evaluating antimicrobial resistance from tulathromycin use in cattle have shown variable results [45,46,47,48,49,50], and there may be scenarios or conditions where resistance is more likely than others. While antimicrobial resistance was not a measured outcome in this trial, the authors are proponents of judicious antimicrobial use and stewardship, and agree that (with other impacts outstanding) reducing antimicrobial use is a favorable outcome. There was a large discrepancy between treatments for AMD on a per animal or per kg HCW basis, with PT being favorable compared to META. The majority of the difference comes from all META cattle receiving tulathromycin at initial processing, while the remainder is explained by differences in the mean count of animals treated for BRD. In higher-risk populations, it is possible the AMD discrepancy between treatments could narrow, but greater antimicrobial use will likely always be expected for cattle that receive metaphylaxis.

Feedlot performance metrics have been shown to decline as cattle are treated more times for BRD [6], which is supportive of overall healthier populations having higher performing animals. It was of critical importance to evaluate trial outcomes on a dead-and-removed-animals-included basis (per animal enrolled), as differences in mortality are better captured, and this basis is the foundation of evaluating economic and emissions outcomes. Improvements in feedlot performance outcomes for META compared to PT were more apparent on a dead-and-removed-animals-included basis. Similar improvements in cattle receiving antimicrobial metaphylaxis have been discussed by Nickell and White [10], and are additionally supported by a meta-analysis of conventional vs. nonconventional beef production practices [51]. While total beef production increased for META cattle compared to PT, feed inputs (DMI) also increased, yielding similar feed efficiency between treatments; DMI also had related implications for economic and greenhouse gas emissions assessments. In addition to improved feedlot performance, META had heavier HCW per animal enrolled compared to PT; this observation is consistent with research evaluating metaphylaxis using macrolides compared to negative controls [40]. No significant differences in other carcass characteristics were observed, which is consistent with Nickell et al. [39] and their comparison of BRD health management strategies using a medium-risk population. However, Tennant et al. [40] reported differences in dressed yield, ribeye area, and empty body fat, but not for quality and yield grades of macrolide-treated cattle compared to negative controls. Generally, cattle treated for BRD, requiring multiple BRD treatments, or with lung lesions at slaughter have presented poorer quality carcasses [40,52,53]; it is plausible that cattle in this trial did not have severe enough BRD incidence to result in carcass quality differences.

Dennis et al. [54] estimated that the beef industry experiences net returns of $532 million to $680 million per year from use of antimicrobial metaphylaxis, and that its elimination could result in surplus losses of $1.81 billion to $2.32 billion for beef producers, and $1.15 billion to $1.47 billion for consumers. Furthermore, Dennis et al. [55] estimated that high-risk feedlot steers received in the winter and administered a “top-tier” metaphylactic antimicrobial (e.g., tulathromycin; [9]) compared to those not receiving metaphylaxis, may be expected to have net returns of $122.55 and $148.65 per animal for those weighing 272 kg (600 lb) and 363 kg (800 lb) on arrival, respectively. These estimates do not agree with results from the economic assessment performed herein. This is likely a function of cattle risk; Dennis et al. [54,55] studies evaluated high-risk cattle classifications, and Nickell and White [10] estimated that for metaphylaxis costs of $18 per animal, expected morbidity may need to be greater than 40% to economically justify metaphylaxis. The mean antimicrobial cost difference between treatments was $21.35 per animal enrolled, which could be similar to Nickell and White’s [10] $18 estimate if inflation-adjusted. Therefore, increased morbidity may have been required to observe treatment differences for estimated net returns. Although META cattle generated more revenue in the partial budget (due to decreased mortality and increased beef production), it appears that because PT cattle had lower costs (namely feed and antimicrobials), resulting differences in net returns were nullified. Notably, the partial budget results indicated that these cattle were not profitable; this was likely due to low fed cattle prices at the time of harvest related to the COVID-19 pandemic, as commercial abattoirs had very little demand for cattle on account of greatly reduced slaughter capacities. An important change that has occurred since the time of the trial (therefore after the framework of prices used in the partial budget) is the emergence of generic alternatives for tulathromycin, resulting in lower prices. At the time of this publication, Draxxin can be purchased for $3.38/mL (ValleyVet.com), and for example, an alternative tulathromycin injection (Increxxa; Elanco Animal Health) can be found for $2.06/mL (ValleyVet.com), which would translate to a metaphylaxis cost of approximately $13.05 per animal enrolled rather than the approximately $25.03 used in the partial budget. Holding other parameters in the economic assessment constant, using lower tulathromycin costs could yield more favorable net returns for META compared to PT. While these economic results provide some insight, a more robust analysis accounting for changes in cattle, feed, and pharmaceutical prices may be warranted as there may be conditions where decisions around BRD health management programs change. An additional consideration could include market timing of finished cattle, as META cattle potentially could have been fed fewer days to achieve similar finished weights as PT, thereby reducing feed, yardage, and interest costs (but also reducing revenue from fed cattle sales); or, PT cattle could have been fed longer to achieve similar finished weights as META, thereby increasing feed, yardage, and interest costs.

Tremendous efforts have been made to improve the environmental sustainability of beef production. Various in-feed technologies have been studied to reduce greenhouse gas emissions in cattle [56,57,58,59], and strategies will likely continue to emerge. Although not specifically targeted for improving greenhouse gas emissions, changes in management practices and use of interventions like antimicrobial metaphylaxis may plausibly have residual effects beyond their original intention. To the author’s knowledge, this is the first publication that attempted to estimate and incorporate production emissions as an outcome for a BRD-related intervention. On a per-animal-enrolled basis, it was estimated that more CO2e was associated with the finishing (feedlot) phase for META cattle compared to PT. This was primarily driven by greater feed consumption by META cattle, which has effects on enteric methane, manure CO2e, and feedlot operation CO2e; as more feed ingredients had to be grown, commodity trucking requirements increased, as did fuel and utilities used for ration production, mixing, and delivery. Thus, even though beef production was greater for META pens than PT, estimated CO2e per unit of final BW or HCW were similar, due to increased finishing CO2e associated with META. On average, the proportion of estimated lifetime CO2e per animal enrolled that occurred pre-feedlot (birth until feedlot procurement) was 76.7% for PT and 76.2% for META. Therefore, the majority of the CO2e associated with each animal during its lifetime was estimated to have occurred prior to feedlot entry. This potentially gives a narrower opportunity for greenhouse-gas-specific interventions that are delivered in the finishing phase to have substantial impacts on the lifetime CO2e of beef animals. However, a major impact that can be had is by simply reducing mortality. A tremendous amount of emissions “cost” is associated with each animal before it enters the feedlot; if an animal dies, those inputs are lost. This is why META cattle had more favorable outcomes for total estimated CO2e per unit of final BW and HCW than PT, as mortality was reduced, and ultimately more beef was produced. This phenomenon should be considered for all animal production strategies that improve production efficiency (not just for metaphylaxis or animal health), as there are likely carryover effects on total carbon footprints. Additionally, altering DOF of the treatment groups to achieve similar final weights could be another consideration that would impact greenhouse gas emissions.

Several limitations of this research should be considered. There was a smaller than planned sample size, for which a more liberal significance threshold was used. With such a sample size, there is increased risk of statistical error. Blinding was not able to be used for the assessment of health outcomes; therefore, there was potential for animal caretakers to increase the rate of animals pulled for treatment in PT pens, knowing that there was no PMI present. While the authors recognize blinding would have been preferable, this mimics a more “real-world” scenario, where health assessors would typically be aware of when cattle are or are not eligible to be treated for BRD (which was the case for all trial and non-trial cattle in these feedlots). There may be some evidence that the impact from lack of animal caretaker blinding was minimal, as the difference in first BRD treatments was approximately proportional to the difference in mortalities between treatments. Misclassification of BRD could have occurred, as animals were treated based only on clinical signs. There are challenges associated with distinguishing BRD from non-BRD respiratory disease (e.g., AIP), and potentially from ruminal acidosis as well. However, it is unlikely that any misclassification that occurred was differential between treatments. Even with a smaller than planned sample size, external validity and overall generalizability of the trial are believed to be relatively strong. Because the research was conducted with steers and heifers, at two distinct locations, and at commercial feedlots mimicking traditional production settings, the results are likely readily applicable to producers procuring similar cattle populations. Morbidity risk in medium-risk feedlot cattle can be variable, and future research should aim to better define this population, and health management approaches. Alternatives to metaphylaxis will continue to emerge and are of critical importance (e.g., strategies to better identify cattle requiring antimicrobial treatment at processing [39]), in particular if similar efficacy can be achieved. Additionally, improving management strategies for cattle populations in cow-calf and stocker sectors to reduce the number of at-risk groups entering feedlots is likely a viable solution to lower antimicrobial reliance [8]. It is recommended that future research use a holistic approach to evaluating outcomes to best assess a broad range of costs and benefits associated with any given intervention.

## 5. Conclusions

The goal of this research was not just to determine a superior BRD management program for medium-risk feedlot cattle, but rather to estimate a comprehensive set of outcomes of high relevance to industry stakeholders by assessing costs and benefits of each program. However, clear differences emerged. Quantified outcomes included morbidity, mortality, antimicrobial use, feedlot performance, carcass characteristics, economics, and greenhouse gas emissions. There are also critical values that are difficult to directly measure that were not discussed in detail, including animal welfare and labor. Certainly, reduced morbidity and mortality would indicate improved animal wellbeing for META cattle compared to PT. Additionally, these reductions may also reduce feedlot labor requirements if fewer cattle require treatment. While using a PT program in lieu of META can be done and result in substantially fewer antimicrobial doses used, there will likely be costs, namely, negative impacts on health, performance, beef production, and greenhouse gas emissions in medium-risk cattle populations. While additional research with similar target populations should be performed, the decision on whether or not to use antimicrobial metaphylaxis will need to be based on the set of outcomes deemed most important. Industry stakeholders including producers, policy makers, and consumers need to evaluate the relative importance of each outcome, i.e., a determination of the degree of antimicrobial use reduction required to justify the numerous costs associated with other values.

## Figures and Tables

**Table 1 vetsci-10-00067-t001:** Dietary ingredient formulations and calculated nutrient composition of starter and finishing rations fed to feedlot steers and heifers at Kansas (KS) and Nebraska (NE) feedlots.

Item	Starter KS	Finishing KS	Starter NE	Finishing NE
Ingredient, % dry matter				
Flaked corn	31.31	68.48	29.97	63.54
Wet distiller grain	22.14	14.95	20.00	15.98
Corn silage	14.12	3.75	7.19	10.07
Alfalfa hay	28.48	-	39.16	2.69
Corn stalks	-	5.20	-	-
Liquid supplement *	3.92	4.50	3.67	4.83
Tallow/corn oil	0.00	3.11	0.00	2.88
Micro-ingredients ^†^	0.03	0.01	0.01	0.01
Nutrient composition				
Dry matter, %	52.14	61.81	56.35	59.38
Crude protein, %	19.33	14.73	17.30	13.86
Fat, %	4.70	7.24	3.98	6.93
Calcium, %	1.00	0.66	1.09	0.83
Phosphorus, %	0.39	0.30	0.49	0.46
Sulfur, %	0.28	0.22	0.30	0.22
NE_m_, Mcal/45.4 kg	83.55	101.63	85.33	108.69
NE_g_, Mcal/45.4 kg	54.90	70.81	54.00	73.63

* A liquid (molasses- and urea-based) protein supplement. ^†^ Feed additives incorporated using Micro-Weigh Systems (Micro Beef Technologies, Amarillo, TX, USA). Finishing diets included 50 g BactaShield (Legacy Animal Nutrition, Wamego, KS, USA), 365 mg monensin (Rumensin; Elanco Animal Health, Greenfield, IN), 75 mg tylosin phosphate (Elanco Animal Health), and for heifers—0.4 mg melengestrol acetate (Zoetis Animal Health, Parsippany, NJ, USA)—per animal daily. For the last 28 to 42 days on feed, 250 mg ractopamine hydrochloride (Zoetis Animal Health, Parsippany, NJ, USA) was delivered per animal daily (except for one KS heifer block).

**Table 2 vetsci-10-00067-t002:** Model adjusted means and standard errors of the mean (SEM) for health and antimicrobial use outcomes of steers and heifers at medium risk for bovine respiratory disease (BRD) that received metaphylaxis (META) or a pull-and-treat (PT) health program for BRD *.

Item	PT	META	*p*-Value
All morbidity ^†^, % (SEM)			
First treatment	18.75 (2.967)	9.44 (1.755)	<0.01
Second treatment	2.89 (1.108)	1.67 (0.686)	0.06
Third treatment	1.13 (0.431)	0.60 (0.273)	0.17
Mean treatment count ^⸙^, *n*/100 animals (SEM)			
BRD	20.26 (3.737)	9.15 (1.805)	<0.01
Non-BRD respiratory disease	0.42 (0.230)	0.78 (0.377)	0.21
Musculoskeletal/trauma	0.64 (0.282)	0.50 (0.237)	0.63
Other	1.20 (0.425)	1.13 (0.405)	0.86
Total	22.45 (4.142)	11.78 (2.273)	<0.01
BRD morbidity, % (SEM)			
First treatment	17.18 (2.927)	7.32 (1.498)	<0.01
Second treatment	2.41 (0.841)	1.24 (0.486)	0.05
Third treatment	0.97 (0.401)	0.52 (0.254)	0.21
BRD AMD ^‡^/animal enrolled, mean (SEM)	0.21 (0.029)	1.10 (0.138)	<0.01
BRD AMD ^‡^/100 kg HCW ^§^, mean (SEM)	0.06 (0.009)	0.30 (0.040)	<0.01
Animals removed from trial, % (SEM)			
BRD	1.05 (0.403)	0.72 (0.303)	0.35
Other	0.55 (0.242)	1.01 (0.366)	0.21
Total	1.55 (0.535)	1.62 (0.555)	0.89
Mortality, % (SEM)			
BRD	0.73 (0.341)	0.44 (0.234)	0.34
Other	1.44 (0.457)	0.56 (0.231)	0.04
Total	2.53 (0.503)	1.15 (0.317)	0.03
Total fallouts ^⸕^, % (SEM)	3.91 (0.953)	2.71 (0.708)	0.10

* Trial was conducted as a randomized complete block design at two feedlots in Kansas and Nebraska, USA, using medium-risk steer or heifer blocks. A total of 1544 heifers and 822 steers were randomly allocated to one of two pens within sex and feedlot to form a block, and pens were randomly assigned to treatment (PT or META). Tulathromycin was administered to all META cattle at initial processing, while PT cattle received tulathromycin for first treatment of clinical BRD. Subsequent antimicrobials used for treatment of clinical BRD were identical in their order and type for both treatments. A total of 16 pens with eight treatment replications were enrolled in the trial. ^†^ All treatment categories combined. ^‡^ AMD = antimicrobial doses. ^§^ HCW = hot carcass weight. ^⸙^ Total animals pulled for treatment within a pen per category; includes pulls from cattle treated multiple times. ^⸕^ Fallouts = all animals removed from the trial plus all mortalities.

**Table 3 vetsci-10-00067-t003:** Model adjusted means and standard errors of the mean (SEM) for feedlot performance outcomes of steers and heifers at medium risk for bovine respiratory disease (BRD) that received metaphylaxis (META) or a pull-and-treat (PT) health program for BRD *.

Item	PT	META	SEM	*p*-Value
Animals enrolled, *n*	1183	1183	-	-
Animals completing trial, *n*	1127	1144	-	-
Initial body weight, kg	261	262	4.4	0.52
Dry matter intake, kg/day	8.88	9.07	0.147	0.06
*Deads and removals excluded* ^†^				
Final body weight ^‡^, kg	586	592	3.9	0.10
Average daily gain, kg	1.48	1.50	0.035	0.13
Gain:feed	0.166	0.166	0.0028	0.83
*Deads and removals included* ^§^				
Final body weight ^‡^, kg	568	583	4.1	0.04
Average daily gain, kg	1.39	1.45	0.045	0.05
Gain:feed	0.157	0.160	0.0039	0.24

* Trial was conducted as a randomized complete block design at two feedlots in Kansas and Nebraska, USA, using medium-risk steer or heifer blocks. A total of 1544 heifers and 822 steers were randomly allocated to one of two pens within sex and feedlot to form a block, and pens were randomly assigned to treatment (PT or META). Tulathromycin was administered to all META cattle at initial processing, while PT cattle received tulathromycin for first treatment of clinical BRD. Subsequent antimicrobials used for treatment of clinical BRD were identical in their order and type for both treatments. A total of 16 pens with eight treatment replications were enrolled in the trial. ^†^ Performance outcomes calculated on the basis of all animals removed from the trial and mortalities excluded from the denominator. ^‡^ A 4% shrink was applied to final pen weights and was accounted for in calculations of average daily gain and gain:feed. ^§^ Performance outcomes calculated on the basis where all animals enrolled in the trial are included in the denominator.

**Table 4 vetsci-10-00067-t004:** Model adjusted means and standard errors of the mean (SEM) for carcass characteristics of steers and heifers at medium risk for bovine respiratory disease (BRD) that received metaphylaxis (META) or a pull-and-treat (PT) health program for BRD *.

Item	PT	META	SEM	*p*-Value
Hot carcass weight, kg/carcass	378	383	3.5	0.12
Hot carcass weight, kg/animal enrolled	362	371	4.0	0.08
Dressed yield, %	64.46	64.59	0.250	0.32
Ribeye area, cm^2^	89.23	90.39	0.910	0.28
12th-rib fat thickness, cm	1.57	1.57	0.033	0.99
Marbling score ^†^	514	519	10.0	0.52
Calculated yield grade	3.18	3.17	0.063	0.87
USDA Yield Grade ^‡^, % (n)	-	-	-	0.80
1	9.32 (105)	9.71 (111)		
2	35.23 (397)	33.86 (387)		
3	38.51 (434)	39.11 (447)		
4	15.00 (169)	15.14 (173)		
5	1.95 (22)	2.19 (25)		
USDA Quality Grade ^‡^, % (n)	-	-	-	0.85
Prime	7.63 (86)	8.75 (100)		
Choice	75.95 (856)	73.58 (841)		
Select and below	16.42 (185)	17.67 (202)		

* Trial was conducted as a randomized complete block design at two feedlots in Kansas and Nebraska, USA, using medium-risk steer or heifer blocks. A total of 1544 heifers and 822 steers were randomly allocated to one of two pens within sex and feedlot to form a block, and pens were randomly assigned to treatment (PT or META). Tulathromycin was administered to all META cattle at initial processing, while PT cattle received tulathromycin for first treatment of clinical BRD. Subsequent antimicrobials used for treatment of clinical BRD were identical in their order and type for both treatments. A total of 16 pens with eight treatment replications were enrolled in the trial. ^†^ Scores ranging from 500 to 599 indicate a modest degree of marbling. ^‡^
*p*-value from multinomial (ordinal) regression testing the null hypothesis that the proportions of carcasses distributed across categories are equal between treatments; values are raw frequency statistics.

**Table 5 vetsci-10-00067-t005:** Model adjusted means and standard errors of the mean (SEM) for economic outcomes from a partial budget assessment of steers and heifers at medium risk for bovine respiratory disease (BRD) that received metaphylaxis (META) or a pull-and-treat (PT) health program for BRD *^†^.

Item	PT	META	SEM	*p*-Value
Purchase cost ^‡^, $/animal enrolled	843.83	845.73	13.752	0.54
Feed and yardage cost ^§^, $/animal enrolled	536.21	550.65	8.743	0.06
Cost of gain ^⸙^, $/animal enrolled	81.54	79.79	1.905	0.24
Cost of BRD antimicrobials ^⸕^, $/animal enrolled	5.83	27.18	0.805	<0.01
Cost of non-BRD treatments ^‖^, $/animal enrolled	0.37	0.42	0.124	0.72
Live sale basis revenue ^⸎^, $/animal shipped	1345.92	1360.79	9.029	0.10
Live sale basis revenue ^⸎^, $/animal enrolled	1290.25	1322.09	12.673	0.07
Dressed sale basis revenue ^⸘^, $/carcass	1344.74	1359.99	12.172	0.21
Dressed sale basis revenue ^⸘^, $/animal enrolled	1289.30	1320.56	16.215	0.12
Net return ^⸓^—live basis, $/animal shipped	−111.43	−113.95	17.532	0.81
Net return ^⸓^—dressed basis, $/carcass	−112.50	−115.66	22.041	0.81
Net return ^⸓^—live basis, $/animal enrolled	−106.40	−110.30	16.519	0.68
Net return ^⸓^—dressed basis, $/animal enrolled	−107.35	−111.83	20.918	0.71

* Trial was conducted as a randomized complete block design at two feedlots in Kansas and Nebraska, USA, using medium-risk steer or heifer blocks. A total of 1544 heifers and 822 steers were randomly allocated to one of two pens within sex and feedlot to form a block, and pens were randomly assigned to treatment (PT or META). Tulathromycin was administered to all META cattle at initial processing, while PT cattle received tulathromycin for first treatment of clinical BRD. Subsequent antimicrobials used for treatment of clinical BRD were identical in their order and type for both treatments. A total of 16 pens with eight treatment replications were enrolled in the trial. ^†^ The partial budget used cattle market prices reflective of the time the trial occurred, October 2019 through July 2020. ^‡^ Feeder steer price = $154.01/45.4 kg; feeder heifer price $138.81/45.4 kg [24]. ^§^ Price includes dry feed, feed markup, and yardage from the Central Plains; $254.14/907 kg [25]. ^⸙^ Feed and yardage cost/45.4 kg live BW gain. ^⸕^ Includes all metaphylactic tulathromycin for META treatment, and all subsequent treatments of clinical BRD for META and PT, including a $1.50 chute charge for each time treated. ^‖^ Cost of non-BRD treatments that were not controlled for in the trial using reflective prices from USDA—NAHMS [26]. ^⸎^ Live fed price for steers = $103.94/45.4 kg; heifers = $104.43/45.4 kg [32]. ^⸘^ The dressed sale base price was the live fed price divided by a common dressed yield of 0.635 (steer price = $163.69/45.4 kg; heifers = $164.46/45.4 kg), then grid adjusted for carcass premiums and discounts [33] based on cattle weight in each category. ^⸓^ Net return = total pen revenue—total pen costs; then divided by the number of cattle shipped or carcasses graded.

**Table 6 vetsci-10-00067-t006:** Model adjusted means and standard errors of the mean (SEM) for calculated estimates of carbon dioxide equivalent (CO_2_e) emissions from the production of steers and heifers at medium risk for bovine respiratory disease (BRD) that received metaphylaxis (META) or a pull-and-treat (PT) health program for BRD *.

Item ^†^	PT	META	SEM	*p*-Value
Pre-feedlot calf footprint ^‡^, kg CO_2_e				
Per animal enrolled	5859.6	5862.4	21.24	0.55
Feedlot finishing footprint ^§^, kg CO_2_e				
Feedlot operations ^⸙^, per animal enrolled	981.0	1005.6	16.73	0.06
Manure ^⸕^, per animal enrolled	402.5	413.0	6.81	0.09
Enteric methane ^‖^, per animal enrolled	399.6	410.4	6.52	0.06
Total finishing ^⸎^, per animal enrolled	1783.1	1829.0	27.63	0.07
Total finishing ^⸎^, per kg final BW	3.18	3.18	0.054	0.94
Total finishing ^⸎^, per kg HCW	4.93	4.92	0.080	0.94
Total footprint ^⸘^, kg CO_2_e				
Per animal enrolled	7642.6	7691.4	33.14	0.08
Per kg final BW	13.66	13.38	0.138	0.09
Per kg HCW	21.20	20.74	0.245	0.10

* Trial was conducted as a randomized complete block design at two feedlots in Kansas and Nebraska, USA, using medium-risk steer or heifer blocks. A total of 1544 heifers and 822 steers were randomly allocated to one of two pens within sex and feedlot to form a block, and pens were randomly assigned to treatment (PT or META). Tulathromycin was administered to all META cattle at initial processing, while PT cattle received tulathromycin for first treatment of clinical BRD. Subsequent antimicrobials used for treatment of clinical BRD were identical in their order and type for both treatments. A total of 16 pens with eight treatment replications were enrolled in the trial. ^†^ BW = shrunk body weight; HCW = hot carcass weight. ^‡^ Calculated estimate of the total CO2e generated from each calf (from cow-calf production until feedlot purchase) based on incoming animal weight and breed. ^§^ Calculated estimates of CO2e generated during the finishing phase (trial period). ⸙ Calculated estimates of CO2e generated from the production of dry feed ingredients, commodity trucking, and fuel and utilities at the feedlot. ^⸕^ Calculated estimates of the total CO2e associated with manure, which includes: manure methane, direct manure N2O, indirect manure N2O from volatilization, and indirect manure N2O from leaching and runoff. ^‖^ Calculated estimates of CO2e enteric methane produced by the animal. ^⸎^ Sum of estimated CO2e generated during the finishing period from feedlot operations, manure, and enteric methane. ⸘ Total calculated estimates of CO2e from the pre-feedlot calf footprint and finishing period footprint.

## Data Availability

Restrictions apply to the availability of these data, and therefore the data are not publicly available; however, data may be shared by the corresponding author upon reasonable request.

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
