# Peer review of "Comprehensive Outcomes Affected by Antimicrobial Metaphylaxis of Feedlot Calves at Medium-Risk for Bovine Respiratory Disease from a Randomized Controlled Trial"

_vetsci, 2023, doi:10.3390/vetsci10020067_

Round 1

Reviewer 1 Report

The manuscript presents detailed data (benefits and disadvantages) of the use of metaphylaxis and "classic therapy" in a selected group of cattle exposed to BRD, pointing to the greater benefits of metaphylaxis, such as generally lower mortality and better average daily gains. However, the current trend, which is the general reduction of the use of antimicrobials in veterinary medicine, should be borne in mind. The study includes many important aspects, such as these related to animal health and feeding, economic assessment or production emissions. The study is useful, especially from the point of view of special group of stakeholders.

Minor comment

Vaccination data could be included in a table for better visualization, e.g. by treatment group and by type and period of administration of the vaccine.

Author Response

Vaccination data could be included in a table for better visualization, e.g. by treatment group and by type and period of administration of the vaccine.

A new appendix table (Table A2) was added to better visualize the differences of processing procedures by blocks.  The table does not include by treatment group, as the procedures were identical between treatment groups (aside from tulathromycin at initial processing) within blocks.

Reviewer 2 Report

Line 36 and 37, my preference would be use antibiotic doses per animal instead of per 100. So 1.1 doses for meta and 0.2 doses for PT. saves me from doing to math in my head as I read to get to real number.

Line 40-41, I disagree with this conclusion and so do your results. If there is no difference in net returns between treatment groups what outcomes indicate that there is substantial costs to not incorporating META?

Line 111-114, need more details about what made these cattle medium risk. As you have stated, medium risk is not well defined, and different veterinarians/managers/buyer could have different definitions. Therefore, for a study such as this you need to define what he age/weight/origin/history/comingling/distance/historicial was so that A) another researcher can replicate your study as close as possible or B) reader can ascertain if this study is relevant to their medium risk cattle definition.

Line 243-245, sentence needs clarification. Can’t use this by itself, this what? This metric?. A lot of which’s in the sentence also.

Line 261-276, Should include something about marketing decisions since you are including carcass data and this information should be reported in results as well. Were cattle fed to set end weight or marketed at same time. In other words did a block of cattle go to harvest on same day or were they marketed when the pen finished. Reality is that carcass characteristics don’t mean much unless we know that cattle were marketed at the same finish. Since you did not report DOF anywhere I am assuming cattle were marketed by block but you need to clarify.

Line 291, This again, should double check manuscript. This (these) has to have an object after it.

Line 299, reword sentence. Using actual prices reported…  should be changed to Actual prices…were used.

Line 309-311, this sentence is pretty repetitive of the previous sentence

Line 312-320, not sure you really need to include these treatments in your partial budget unless there is really a reason, which would be an important determination. In theory non-BRD costs should be the same between treatments so they are a non-factor in the analysis. You should only include relevant numbers in the partial budget. Especially since you are just using industry averages, instead of real numbers from study cattle. Unless you think there is some reason there would be a correlation between META/PT and non-BRD cases.

Line 362-366, pre feedlot emission estimates seem like they have little value. Making assumptions about estimates you have no control over seems pretty pointless.

Line 396, does random effect include feedlot or only block? Would need to account for feedlot effect since different geographic locations, pen riders, feed delivery, etc.

Line 430, just to make sure there is no confusion I would clarify that metaphylaxis treatment is not considered to be a first-time treatment

Line501-503, see comment above from methodology, pre-feedlot emissions seems pretty pointless.

Line 520, table 6, these numbers are not making sense to me, check your calculations. For the feedlot phase the C02 per kg final body weight is the same. When you go to total footprint, since META calves started with slightly bigger pre-feedlot C02 and their feedlot C02 is higher, how does there lifetime CO2 per final BW go down? Your denominator final BW kg shouldn’t have changed and numerator increase?

Line 522-524, my bias, but outcomes research approach is just telling me you threw a bunch of stuff at the wall to see what would stick, instead of having targeted study.

Line 592-595, not sure value of this sentence

Line 632-636, you should also state that your study design severely limited partial budget analysis. Your two biggest cost categories purchase price and DOF were the same since they were bought and marketed (assuming since you did not report DOF) on the same days. So the only potential difference you have is some slight increase in DMI and ADG, and treatment expense. Theoretically, if META cattle could have been marketed a few days sooner you would have reduced feed expense leading to a real difference in your partial budget. This design is a limitation to your economic as well as your emission analysis and should be noted.

Line 649-653, understand wanting to point this out but by the time this is published your updated prices will no longer be accurate either which is underlying problem with including budget data in a study. You are better off making a statement that partial budgets need to be adjusted as prices associated with feeder cattle, feed, processing and fat cattle change.

Author Response

Line 36 and 37, my preference would be use antibiotic doses per animal instead of per 100. So 1.1 doses for meta and 0.2 doses for PT. saves me from doing to math in my head as I read to get to real number.

This change was made.

Line 40-41, I disagree with this conclusion and so do your results. If there is no difference in net returns between treatment groups what outcomes indicate that there is substantial costs to not incorporating META?

We believe the reviewer confused “costs” as referring to only economics, whereas we were meaning to refer to costs for other outcomes (e.g., performance, health, emissions).  The sentence was reworded to avoid this confusion.

Line 111-114, need more details about what made these cattle medium risk. As you have stated, medium risk is not well defined, and different veterinarians/managers/buyer could have different definitions. Therefore, for a study such as this you need to define what he age/weight/origin/history/comingling/distance/historicial was so that A) another researcher can replicate your study as close as possible or B) reader can ascertain if this study is relevant to their medium risk cattle definition.

Cattle weight is listed earlier in the paragraph.  Estimated age and mean transportation distance was added.  The other components (sale-barn origin, commingling, unknown health history) listed as-is accurately describe the cattle population.  As a side note, the greatest influencer for the personnel/group making the risk classification is history with similar cattle purchased from the same locations and season.

Line 243-245, sentence needs clarification. Can’t use this by itself, this what? This metric?. A lot of which’s in the sentence also.

These sentences were rewritten for clarity.

Line 261-276, Should include something about marketing decisions since you are including carcass data and this information should be reported in results as well. Were cattle fed to set end weight or marketed at same time. In other words did a block of cattle go to harvest on same day or were they marketed when the pen finished. Reality is that carcass characteristics don’t mean much unless we know that cattle were marketed at the same finish. Since you did not report DOF anywhere I am assuming cattle were marketed by block but you need to clarify.

This was a good catch by the reviewer.  Both treatments (pens) within a block were harvested on the same day (i.e., same days-on-feed).  This detail was added into this section as well as the mean (and range) days-on-feed.

Line 291, This again, should double check manuscript. This (these) has to have an object after it.

This correction was made.

Line 299, reword sentence. Using actual prices reported…  should be changed to Actual prices…were used.

This correction was made.

Line 309-311, this sentence is pretty repetitive of the previous sentence

Agreed, but we believe it is important to distinguish (and reiterate) the difference between PT and META, i.e., the differences of when each antimicrobial was used for each BRD treatment.

Line 312-320, not sure you really need to include these treatments in your partial budget unless there is really a reason, which would be an important determination. In theory non-BRD costs should be the same between treatments so they are a non-factor in the analysis. You should only include relevant numbers in the partial budget. Especially since you are just using industry averages, instead of real numbers from study cattle. Unless you think there is some reason there would be a correlation between META/PT and non-BRD cases.

We agree with the reviewer that we could have justified not accounting for non-BRD morbidity, due to the lack of significant differences between treatments.  It likely does not impact the results either way to have it included.  However, along the lines of the last point made in this comment, we did hypothesize in the third paragraph of the discussion (Section 4) that antimicrobial use targeting BRD, may have impacts on health outcomes beyond BRD (i.e., we do not believe tulathromycin is mutually exclusive of manufacturer label claims).  There is some potential evidence of this as we observed a difference in non-BRD mortality.  Finally, by accounting for non-BRD morbidity, it perhaps helps to provide more realistic final estimates for cost and net return of the pens, regardless of whether or not it was impacted by treatment.

Line 362-366, pre feedlot emission estimates seem like they have little value. Making assumptions about estimates you have no control over seems pretty pointless.

We disagree with this point, reasons for why it was deemed critical to include pre-feedlot estimates are discussed in lines 685 through 695.

Line 396, does random effect include feedlot or only block? Would need to account for feedlot effect since different geographic locations, pen riders, feed delivery, etc.

The random intercept as it was specified accounts for both the clustering of pens within blocks and blocks within feedlots (i.e., yes, the differences between feedlots were accounted for).

Line 430, just to make sure there is no confusion I would clarify that metaphylaxis treatment is not considered to be a first-time treatment

A statement to reiterate this was added at the beginning of Section 3.1 (lines 434-436).

Line501-503, see comment above from methodology, pre-feedlot emissions seems pretty pointless.

See comment above for rationale in the discussion.

Line 520, table 6, these numbers are not making sense to me, check your calculations. For the feedlot phase the C02 per kg final body weight is the same. When you go to total footprint, since META calves started with slightly bigger pre-feedlot C02 and their feedlot C02 is higher, how does there lifetime CO2 per final BW go down? Your denominator final BW kg shouldn’t have changed and numerator increase?

We understand the confusion, however the numbers are correct.  Comparing total finishing emissions to the total footprint (lifetime) emissions per animal enrolled, the percent difference between treatments is much smaller for the total footprint than total finishing.  Therefore, the difference between the numerators is smaller, but the denominator stays the same (and is larger for META than PT; Tables 3 and 4).  You can do the divisions directly in the tables to see that the numbers align (keeping in mind that the result won’t be the exact same as these are model adjusted means, and means of ratios are not the same as ratios of means, but it will be close). 

For example, PT total finishing emissions per animal enrolled = 1,783.1 kg CO2e; 1,783.1 kg CO2e / 568 kg final BW = 3.14 kg CO2e/kg final BW.  The PT total footprint emissions per animal enrolled = 7,642.6 kg CO2e; 7,642.6 kg CO2e / 568 kg final BW = 13.45 kg CO2e/kg final BW. 

The META total finishing emissions per animal enrolled = 1,829 kg CO2e; 1,829 kg CO2e / 583 kg final BW = 3.14 kg CO2e /kg final BW.  The META total footprint emissions per animal enrolled = 7,691.4 kg CO2e; 7,691.4 kg CO2e / 583 kg final BW = 13.19 kg CO2e/kg final BW. 

Another important consideration in these calculations (which was mentioned in the discussion) is that all calculations are done on a dead and removed animals included basis; when considering the total footprint, the cost of losing more animals (mortalities and removals) in the PT treatment than META (manifesting as additional live or carcass weight) is what is driving the difference.

Line 522-524, my bias, but outcomes research approach is just telling me you threw a bunch of stuff at the wall to see what would stick, instead of having targeted study.

The authors aren’t sure if this is just a flippant comment or requires a response. We would argue that it was a targeted study, specifically for addressing important outcomes of relevance to a variety of stakeholders and decision-makers (which fits, by definition, an outcomes research approach as per the scientific literature on this topic).

Line 592-595, not sure value of this sentence

The sentence was edited, but we would prefer to retain it’s overall message, as it references several papers that have evaluated antimicrobial resistance from tulathromycin.  Across the papers, there were cases where resistance was and was not evident, meaning that it may not always be a certainty, and there are likely conditions where resistance is more likely than others.  As we did not evaluate antimicrobial resistance, we believe this is an important observation as the level of resistance that occurred in this trial is unknown.

Line 632-636, you should also state that your study design severely limited partial budget analysis. Your two biggest cost categories purchase price and DOF were the same since they were bought and marketed (assuming since you did not report DOF) on the same days. So the only potential difference you have is some slight increase in DMI and ADG, and treatment expense. Theoretically, if META cattle could have been marketed a few days sooner you would have reduced feed expense leading to a real difference in your partial budget. This design is a limitation to your economic as well as your emission analysis and should be noted.

These points were added and discussed in lines 661-669 for the economic analysis, and 698-700 for the emission analysis.  However, we disagree that having a fixed/equal purchase price and DOF is an experimental design limitation.

Line 649-653, understand wanting to point this out but by the time this is published your updated prices will no longer be accurate either which is underlying problem with including budget data in a study. You are better off making a statement that partial budgets need to be adjusted as prices associated with feeder cattle, feed, processing and fat cattle change.

See response to the previous comment directly above.

Round 2

Reviewer 2 Report

authors have adequately addressed my comments